# Permeability of Metformin across an In Vitro Blood–Brain Barrier Model during Normoxia and Oxygen-Glucose Deprivation Conditions: Role of Organic Cation Transporters (Octs)

**DOI:** 10.3390/pharmaceutics15051357

**Published:** 2023-04-28

**Authors:** Sejal Sharma, Yong Zhang, Khondker Ayesha Akter, Saeideh Nozohouri, Sabrina Rahman Archie, Dhavalkumar Patel, Heidi Villalba, Thomas Abbruscato

**Affiliations:** 1Department of Pharmaceutical Sciences, Jerry H. Hodge School of Pharmacy, Texas Tech University Health Sciences Center, Amarillo, TX 79106, USA; 2Center for Blood-Brain Barrier Research, Jerry H. Hodge School of Pharmacy, Texas Tech University Health Sciences Center, Amarillo, TX 79106, USA

**Keywords:** metformin, ischemic stroke, repurposing, transporters, in vitro, BBB, co-culture, P-GP, Octs, permeability

## Abstract

Our lab previously established that metformin, a first-line type two diabetes treatment, activates the Nrf2 pathway and improves post-stroke recovery. Metformin’s brain permeability value and potential interaction with blood–brain barrier (BBB) uptake and efflux transporters are currently unknown. Metformin has been shown to be a substrate of organic cationic transporters (Octs) in the liver and kidneys. Brain endothelial cells at the BBB have been shown to express Octs; thus, we hypothesize that metformin uses Octs for its transport across the BBB. We used a co-culture model of brain endothelial cells and primary astrocytes as an in vitro BBB model to conduct permeability studies during normoxia and hypoxia using oxygen–glucose deprivation (OGD) conditions. Metformin was quantified using a highly sensitive LC-MS/MS method. We further checked Octs protein expression using Western blot analysis. Lastly, we completed a plasma glycoprotein (P-GP) efflux assay. Our results showed that metformin is a highly permeable molecule, uses Oct1 for its transport, and does not interact with P-GP. During OGD, we found alterations in Oct1 expression and increased permeability for metformin. Additionally, we showed that selective transport is a key determinant of metformin’s permeability during OGD, thus, providing a novel target for improving ischemic drug delivery.

## 1. Introduction

Metformin (MF) is a commonly prescribed, first-line therapy for treating type two diabetes mellitus [1]. The United States Food and Drug Administration (US-FDA) approved it almost thirty years ago and it has reported no major adverse effects [2]. MF reduces blood glucose levels by inhibiting peripheral and hepatic glucose production without affecting insulin sensitivity, therefore not causing hypoglycemia [3]. Outside of its application in treating diabetes, increasing numbers of pre-clinical and clinical studies show that pre- and post-treatment of the drug has a protective effect against various neurological disorders and ischemic stroke [4,5,6,7,8,9]. Stroke is the fifth leading cause of death in the US and about 87% of all strokes are ischemic strokes [10]. The most common cause of ischemic stroke is a partial or complete blockage of blood flow to the brain due to occlusion of a blood vessel from a clot, resulting in neurological loss or even death [11].

There is a clear unmet clinical need for new and better treatments for CNS diseases and ischemic stroke. Tissue plasminogen activator (tPA) is the only US-FDA-approved drug treatment for ischemic stroke, whose efficacy is compromised if not administered within 3–4 h of stroke onset [12]. Another approved intervention, mechanical or endovascular thrombectomy, is used to treat large vessel occlusion ischemia; however, the procedure is contraindicated in patients with a high risk of intracranial or systemic hemorrhage [13]. Over the last two to three decades, only a few potential CNS drugs have had the probability of getting beyond phase three clinical trials for ischemic stroke [14]. Drug repurposing or repositioning is a process that involves identifying a new indication for an approved or investigational drug that would be outside the original medical indication, with the purpose of accelerating the development of an effective therapy indication [15]. Therefore, repurposing a safe, effective, and established therapeutic such as MF could provide a much faster bench-to-patient pharmacological transition. 

One underlying mechanism of MF as a neurotherapeutic is that it activates AMP-activated protein kinase (AMPK), causing downstream activation of signaling pathways such as NF-κB and mTOR that lead to reduced inflammatory and oxidative-stress responses in the brain [16,17,18]. Previously published data from our group also strongly suggests that MF activates nuclear factor erythroid 2-related factor (Nrf2) in the brain of stroke animals [19]. We showed that MF counteracts the cerebrovascular toxicity caused by tobacco smoking and electronic cigarette vaping (known risk factors for ischemic stroke) by protecting the blood–brain barrier’s (BBB) integrity in stroke-subjected animals [8]. Another essential aspect for determining MF’s neuroprotection is understanding its blood-to-brain permeability and potential interaction with uptake and efflux transporters present at the BBB during normal physiological conditions or normoxia and diseased states, such as ischemic stroke.

MF is a small hydrophilic molecule (logD −6.13) with a molecular weight of 129 Dalton that is positively charged (pKa 12.4) at physiological pH [20,21]. These physicochemical properties suggest that MF permeation across biological membranes does not occur through rapid passive diffusion unless its transport is driven either by membrane potential (carrier-mediated transport) or simply by paracellular diffusion (between the cellular gaps). The BBB mainly comprises brain endothelial cells that closely interact with supporting cells such as astrocytes, microglia, pericytes, and neurons to induce and maintain integrity and function [22,23,24]. Additionally, tight junctional proteins, such as occludin, claudin-5, and ZO-1, expressed by endothelial cells, are crucial for forming a paracellular seal to restrict the movement of molecules or drugs across the BBB [25,26]. This restriction by the BBB stops at least 95% of investigational molecules in the process of drug development [27]. The endothelial cells of the BBB express uptake transporters, solute carriers (SLCs) subtypes of organic cation transporters (Octs) such as Oct1 (SLC22A1), Oct2 (SLC22A2), and Oct3 (SLC22A3) for the transport of hydrophilic cationic CNS drugs into the brain [14]. Octs have been shown to interact with endogenous substrates such as monoamine neurotransmitters and with CNS drugs such as antivirals, tricyclic anti-depressants, and the anti-diabetic MF [28,29,30]. MF has been well known to interact with Octs in organs such as the liver and kidney [31,32,33]. 

Our lab has previously explored the contribution of organic anionic transporters (OATPs) in the transport of a potent opioid receptor agonist, biphalin, during hypoxia and further established an initial time window for brain entry [34]. Another study showed that the functional expression OATP1A2 (an isoform of OATPs) is crucial for the therapeutic efficacy of the statins during stroke recovery [35]. More recently, it was reported that Oct1 and Oct2 transporter expression variations during in vivo stroke conditions lead to enhanced permeability of memantine [36]. Furthermore, it was shown that the transporter-mediated permeation of the drug was increased by two fold and was critical to achieving therapeutic efficacy. Another study showed that expression of Octs in an in vitro human BBB is required to transport carnitine for neuronal homeostasis [37]. Thus, it can be implied that endogenous transporters at the BBB play a vital role and provide a novel approach for the development of CNS drug delivery during ischemic stroke. These reports further warrant evaluation of the permeability of a drug candidate, such as MF, which is expected to use carrier-mediated transport (as it has been shown to be a substrate of Octs in other organs) to gain access to the brain during normal and pathophysiological situations.

Besides uptake transporters, efflux transporters such as plasma glycoproteins (P-GP) are one of the major efflux transporters at the BBB that restrict brain entry [38,39,40]. P-GP restricts solutes or toxins from entering the brain parenchyma from blood circulation and their interaction with MF at the BBB is currently unknown. In the present study, for the first time, we examined the permeability of MF during normal physiological conditions or normoxia and oxygen glucose-deprived (OGD) conditions. We determined that MF uses a saturable mechanism for its transport across an in vitro co-culture model of BBB using a co-culture of bEnd.3 cells (immortalized mouse brain endothelial cells) and mouse primary astrocytes. In addition, we show an alteration in the expression of Oct1 in bEnd.3 cells during OGD time points and enhanced MF permeation into the brain, during which carrier-mediated transport plays a crucial role in the drug’s transport. Lastly, we found that MF does not interact with one of the major efflux proteins, P-GP, in P-GP overexpressing cells.

## 2. Materials and Methods

### 2.1. Cell Culture and bEnd.3/Astrocyte Co-Culture

bEnd.3 cells passages 21–24 (ATCC, Manassas, VA, USA) were cultured in Dulbecco’s modified Eagle’s medium (Sigma, St. Louis, MO, USA) supplemented by 10% FBS (Atlanta Biologicals, Minneapolis, MN, USA) and 1% each of non-essential amino acid and penicillin/streptomycin (PS) solution (Sigma Aldrich). Cells were then maintained in a humidified cell culture incubator at 37 °C and with 5% CO_2_/95% air. Mouse primary astrocytes were obtained from the cerebral cortices of one-day-old CD-1 mouse pups (Charles Rivers Laboratory) according to the previously published methods [41]. After isolating the brain, cerebral cortices were isolated, meninges were removed, and cortices free from meninges were placed in Hanks’ balanced salt solution (HBSS) without calcium and magnesium, supplemented with gentamycin (10 µg/mL). Then, cortices were digested with 0.25% trypsin for 10–15 min at 37 °C, followed by neutralizing with FBS containing Dulbecco’s modified Eagle’s medium containing 10% FBS and 1% PS solution. The cells were then seeded into a cell culture T75 flask and the medium was refreshed every 3 days until reaching confluency. 

For bEnd.3 and astrocyte co-culture, the transwell inserts (0.4–1 µm pore size, 12-well; Corning, Lowell, MA, USA) were inverted and astrocytes at a density of 150,000 cells per insert were seeded onto the basolateral side of the insert membrane and were allowed to adhere for 4 h. The transwell inserts were then inverted back and astrocytes were allowed to grow for 2 more days in the astrocyte medium. Then, bEnd.3 cells with a density of 50,000 cells per insert were seeded onto the apical side of the inserts. The co-culture of primary astrocytes and bEnd.3 cells was grown for 8 more days by changing the media for both cells every other day. Permeability experiments were completed on days 8–10 from day 1 of the co-culture establishment.

### 2.2. Barrier Integrity Measurements

The integrity of the co-culture setup was carried out using two techniques: (1) Through measurement of trans-endothelial cell resistance (TEER) of the transwell inserts and (2) by permeability assessment (apical to basolateral) of sodium fluorescein (NaF) (Sigma Aldrich), a low molecular weight BBB marker, according to previously published method [42]. All permeability experiments were conducted with the co-cultured transwell inserts that had TEER value measurements of >70 Ω·cm^2^. The TEER was measured by EVOM resistance meter (World Precision Instruments, Sarasota, FL, USA) using the STX-2 electrodes. For NaF permeability experiments, the media was removed from both of the compartments (apical and basolateral) and rinsed with HBSS buffer and incubated at 37 °C for 30 min. Then, 500 μL of 10 μg/mL of NaF in HBSS was added to the apical chamber of the inserts and 100 μL was collected from the basolateral compartment of the wells in duplicates for concentration determination. The collected volumes were replaced with 200 μL of HBSS buffer to avoid the back diffusion of NaF. The collection was performed at time points 30, 60, and 120 min. The amount of NaF was measured at an absorbance using a fluorescent microplate reader (PerkinElmer, Waltham, MA, USA) with excitation and emission wavelengths of 485 and 520 nm. Then, the permeability coefficient (PC, in cm/min) was determined using the following equation:(1)$$\mathrm{PC}=\frac{\mathrm{dQ}}{\mathrm{dT}}\times \frac{1}{C_{0}\times A}$$
where dQ/dt is the diffusion rate of NaF across the membrane, A is the area of the transwell insert, and C_0_ is the initial concentration of the NaF added in the donor compartment. The blank transwell insert (without cells) was added to the experimental study group to negate its PC value because blank inserts themselves provide resistance to the buffer and compound:(2)$$\frac{1}{{\mathrm{PC}}_{\mathrm{cells}}}=\frac{1}{{\mathrm{PC}}_{\mathrm{total}}}-\frac{1}{{\mathrm{PC}}_{\mathrm{blank}}}$$

The PC value of NaF (MW 376 Da) was compared with another low molecular weight BBB marker, [^14^C] sucrose (MW 342 Da), used previously in our lab. We found no statistically significant difference between them (Appendix A). 

### 2.3. [^14^C] MF Permeability a Using Self-Inhibition Study

Permeability experiments for [^14^C] MF (Moravek Biochemicals, Brea, CA, USA) were conducted similarly to as explained above for NaF. 0.15 µCi/mL or 10 µM of [^14^C] MF was chosen as a pharmacologically relevant concentration because of the observed steady-state concentration of 1.8 µg/mL or 13 µM in human subjects. Then, MF was diluted in HEPES (in mM 120 NaCl, 1 CaCl_2_, 25 HEPES, 1 KH_2_PO_4_, 2 KCl, 1 MgSO_4_, 10 D-glucose) buffer and added to the apical chamber of the transwell inserts containing pre-incubated concentrations of unlabeled MF (Sigma Aldrich) at 1 mM, 10 mM, and 20 mM. 100 μL of samples of [^14^C] MF were collected from the basolateral chamber at time points 5, 15, 30, 60, and 120 min. The radioactivity of the collected samples was evaluated using a liquid scintillation counter (Beckman Coulter, Brea, CA, USA), and the permeability value for [^14^C] MF in the presence and absence of self-inhibitory concentrations was determined using Equations (1) and (2).

### 2.4. MF Permeability Using Transporter-Specific Inhibitors

Mitoxantrone (Sigma Aldrich), a known Oct1 inhibitor at 25 µM, and corticosterone (Sigma Aldrich), a known Oct1, 2, and 3 inhibitors at 150 µM, were preincubated for 30 min at the apical and basolateral chambers of transwell inserts prior to and during the transport experiments for 120 min. The transport experiment was initiated by adding 500 μL of 10 µM MF at the apical chambers and samples were collected from each group at 5, 15, 30, 60, and 120 min time points. Quantification was conducted through a highly sensitive LC-MS/MS method. After quantification, the permeability value for MF in the presence and absence of transporter-specific inhibitors was determined.

### 2.5. MTS Cell Viability Assay

Cell viability was measured by MTS (dimethylthiazol carboxymethoxyphenyl sulfophenyl tetrazolium) assay using the CellTiter 96 kit Aqueous assay (Promega) according to the provided manufacturer’s instructions. The assay was conducted to ensure that the concentration of MF and inhibitors used in permeability studies is not toxic to the cells. bEnd.3 cells and primary astrocytes were seeded in separate 96 wells plates. After confluency, plates were incubated with different concentrations of unlabeled metformin (100 µM, 1 mM, 10 mM and 20 mM) and Octs inhibitors for 3 h. After incubation, the absorbance wavelength of formazan was read at 490 nm using fluorescent microplate reader. The values were calculated as a percentage of the control value.

### 2.6. LC-MS/MS Method Development, Sample Preparation and Analysis

HBSS buffer was spiked with MF to form the stock solution to achieve a calibration curve range of 3.9 ng/mL to 2000 ng/mL. In addition, two separate blank control solutions, one of them spiked with internal standard, were considered for the study. Fifty μL of the unknown samples obtained from transporter-specific inhibition, oxygen–glucose deprivation (explained below in Section 2.6), and p-GP efflux (explained below in Section 2.8) studies were spiked with 20 μL of 500 ng/mL of internal standard Metformin-D6 hydrochloride (Cayman Chemical Company). Then, the mixture was vortexed for 5 min, after which 930 μL of 2 mM of Ammonium Acetate (Sigma Aldrich) solution was added to the mixture. The solution was further vortexed for 5 min and was centrifuged at 10,000× *g* rpm for 5 min at 4 °C. Then, 500 μL of the supernatant was inserted in deep well plates (Waters Corporation, Milford, MA, USA) and analyzed using the LC-MS/MS method. The ultra-high performance liquid chromatography (UHPLC) system (Shimazdu Corporation, Columbia, MD, USA) equipped with AB SCIEX QTRAP 5500 triple quadrupole mass spectrometer (Foster City, CA, USA) was used. The data acquisition and analysis were performed using Analyst software 1.7.0.

Chromatographic separations were performed using an XBridge BEH C18 sorbent (50 × 3.0 mm, 3.5 μm, Waters Corporation, Milford, MA, USA). The mobile phase A was used as 2 mM ammonium acetate in 5% Acetonitrile (Fisher Scientific, Waltham, MA, USA) in LCMS water and the mobile phase B was used as 100% Acetonitrile. LCMS grade water (Fisher Scientific) was used as rinsing solution. The gradients separation was set up as 100% of A for 0 to 1 min, 90% of B at 1.2 min, and held until 2.5 min. At 2.6 min, the percentage of gradient was brought back to initial concentration of 100% of A and held until 4 min of total runtime. The flow rate was set at 0.35 mL/min, the column temperature was maintained at 45 °C, and injection volume was set at 2 µL. The retention time of MF and its internal standard were ~0.9 min. Multiple reaction monitoring was conducted in negative mode and the transitions for MF and internal standard were 130.1 → 60.1 *m/z* and 136.1→ 60.1 *m/z*, respectively. The collision energy for MF and IS was 20 and 20 volts, respectively.

### 2.7. In Vitro Hypoxia Modelling: Oxygen Glucose Deprivation (OGD)

The cells were exposed to in vitro hypoxia model by OGD, according to previously explained protocol [43,44]. In brief, the co-cultured transwell inserts were exposed to OGD for 2, 4, and 6 h for permeability experiments. In another set of study, bEnd.3 cells and primary astrocytes grown on petri dishes (Corning, Lowell, MA, USA) were exposed to OGD for 2, 4 and 6 h to check the expression of Oct1, 2, and 3. For the experiment, media was removed and replaced with media bubbled with 95% N2/5% CO_2_. Dulbecco’s phosphate buffered saline solution and glucose-free Earle’s balanced salt solution (in mM 140 NaCl, 0.83 MgSO_4_, 5.36 KCl, 1.02 NaH_2_PO_4_, 1.18 CaCl, 26.19 NaHCO_3_, adjusted pH 7.4) to achieve an aglycemic condition. Then, the cells were transferred to custom-made hypoxic chamber (Coy Laboraties, Grasslake, MI, USA) with 95% N_2_ and 5% CO_2_ at 37 °C to induce hypoxia (1% oxygen).

### 2.8. Western Blot for Octs Expression

Caco-2 cells (ATCC, Manassas, VA, USA) was used as a positive control because they have shown to express Octs [21]. After OGD time points, the bEnd.3 cells and primary astrocytes were lysed using radioimmunoprecipitation assay buffer (RIPA) supplemented with proteinase and phosphatase inhibitor cocktails and then collected for estimation of protein concentration using a BCA assay. After solubilizing samples using Laemmli buffer, 30 μg of protein was loaded per well separated by 10% Tris–glycine polyacrylamide precast gel (Bio-Rad Laboratories, Hercules, CA, USA) and then transferred to a polyvinylidene fluoride (PVDF) membrane (Thermo Fisher, Waltham, MA, USA) for 2 h at 90 V. Then, the membrane was incubated in blocking solution containing 5% bovine serum albumin to block nonspecific binding for 2 h at room temperature. Then, the membranes were immunoblotted with the appropriate primary antibodies overnight at 4 °C with the following dilutions: rabbit monoclonal anti-Oct-1 (1:1000) antibody (Cell Signaling Technology, Danvers, MA, USA), rabbit monoclonal anti-Oct-2 (1:1000) (Abcam) and rabbit polyclonal anti-Oct-3 (1:500) and mouse monoclonal anti-beta actin antibody (1: 10,000) (Sigma Aldrich) in TBST with 5% bovine serum albumin. After 3 times washing with TBST for 10 min each, membranes were incubated with anti-rabbit (Sigma Aldrich) or anti-mouse (Sigma Aldrich) IgG-horseradish peroxidase secondary antibody (1:10,000) in TBST with 5% bovine serum albumin for 2 h at room temperature. After 3 times of 10 min wash with TBST, the protein signals were detected using enhanced chemiluminescence substrate (Westernsure chemiluminescent substrate) and visualized in LI-COR; C-Digit blot scanner. The protein bands were quantified relative to beta-actin in Image J 1.53t software and are reported as ratios of the control group.

### 2.9. P-Glycoprotein Efflux Transporter Substrate Assay

For P-GP efflux assay, MDR1-MDCK 12-well transport assay (MB Biosciences LLC, Natick, MA, USA) was used. The cells were maintained in high glucose DMEM media supplied with 10% FBS, non-essential amino acid, penicillin/streptomycin, *L*-glutamine, and colchicine as recommended by the supplier. After the shipping medium was changed to a fresh MDR1-MDCK cell culture medium, the plate was kept in the incubator for 24 h before performing the experiment. 10 µM of MF in HBSS was added to the apical compartment of the transwell inserts in presence and absence of a potent p-GP inhibitor, Cyclosporine A (CsA) at 10 µM concentration (Sigma, St. Louis, MO, USA). The inhibitor was present on both sides of the compartments during pre-incubation of 30 min and during the transport experiment. After 120 min of incubation at 37 °C, samples were collected from the basolateral chamber and quantified by LC-MS/MS method. For groups in which MF was added in the basolateral compartment, samples were collected from the apical chamber after 120 min. The permeability coefficient (PC, cm/s) was calculated using the published literature method. The PC value from apical (A) to basolateral (B) compartment in the presence and absence of CsA was used to calculate the unidirectional flux ratio (UFR), which is defined by the following equation:(3)$$UFR=\frac{PC,\;A-to-B\left(+CsA\right)}{PC,A-to-B\left(-CsA\right)}$$

In a parallel study, MF was added to the basolateral compartment and samples were collected from the apical chamber to evaluate the B-to-A PC value, which allows calculating the efflux ratio (ER), defined by the following equation:(4)$$ER=\frac{PC,\;B-to-A}{PC,A-to-B}$$

### 2.10. Data Analysis

All values are presented as mean ± SD. Unpaired student’s *t*-test was used to compare two groups. The comparison of three or more than three groups was conducted by one-way analysis of variance (ANOVA) followed by Tukey’s post hoc multiple comparison test (Prism, version 9.0; GraphPad Software Inc., San Diego, CA, USA). *p* values of <0.05 were considered to be statistically significant.

## 3. Results and Discussion

### 3.1. MF Is Highly Permeable and Uses Saturable Transport for Its Movement across In Vitro Co-Culture Model of BBB

A desirable PC value for a compound is one of the major prerequisites in vitro ADME properties for a potential CNS drug [45,46]. According to published studies, an in vitro PC value of >6 × 10^−4^ cm/min is considered a highly permeable drug across the BBB. Our study found MF’s PC value to be 24.5 ± 3.14 × 10^−4^ cm/min, which is ~four fold greater than the high PC value threshold (Figure 1C). Additionally, we compared the PC value of MF with compound **11a**, a neurolysin activator with improved BBB permeability (11.4 × 10^−4^ cm/min), as shown previously by our lab and collaborators [47]. We found a two fold increase in permeability for MF (Figure 1B). Notably, the permeability experiments for MF and compound **11a** were conducted on a similar co-culture setup in our lab. 

Next, to explore whether the mechanism behind MF’s transport across the in vitro BBB model is saturable, we measured the PC value of [^14^C] MF as a tracer in the presence of an increasing concentration of cold MF at 1 mM, 10 mM, and 20 mM. We also performed MTS cell viability assay to show that these concentrations are not toxic to both bEnd.3 cells and primary astrocytes (Appendix A). As shown in Figure 1C, with a self-inhibitory concentration of 1 mM, there is no significant decrease in the permeability of [^14^C] MF (21.17 ± 2.96 × 10^−4^ cm/min). However, with a higher inhibitory concentration of 10 mM, there was a significant decrease in [^14^C] MF transport (9.4 ± 3.98 × 10^−4^ cm/min, *p* < 0.01), with no further reduction at 20 mM (9.9 ± 3.76 × 10^−4^ cm/min), indicating the saturation of transporters. Previously, it was reported that the apparent IC_50_ value of MF for uptake of tracer [^14^C] MF was around ~2 mM [48]. Consistent with this previously reported value, we found the saturation concentration was above 1 mM in our study. Furthermore, it can be extrapolated from Figure 1C that ~35% of MF’s transport is active transporter-mediated (the inhibition component of the transport from 28.15 ± 7.70 × 10^−4^ cm/min to 9.9 ± 3.76 × 10^−4^ cm/min is active transport). In comparison, the rest, ~65%, is suggested to be due to paracellular passive diffusion of the molecule across the in vitro BBB co-culture model.

### 3.2. Role of Organic Cation Transporters (Octs) for MF Permeability across the BBB

Studies have shown that brain endothelial cells and brain microvessels have protein and mRNA expressions of Octs [16]. Studies reported the expression of Oct1 and Oct2 in cultured brain endothelial murine and human cell lines [49,50]. MF has been shown to be a substrate for Oct1 in the liver and Oct2 in the kidneys [51,52]. Additionally, studies have shown that MF undergoes saturable uptake in single transporter-expressing cell lines for Oct1, Oct2, and Oct3 with apparent Km values of 3.1, 0.6, and 2.6 mM, respectively [53,54,55]. Moreover, the 10 µm MF concentration used in our transport assays is within the reported Km values of the Oct1, Oct2, and Oct3, suggesting that these transporters are not saturated at this concentration [53]. A recently published study showed targeting Oct1 and Oct2 provided efficient pharmacotherapy for ischemic stroke treatment using the cationic drug memantine [36]. Thus, we assume exploring Octs for MF will give new insight into MF’s therapeutic utility as a potential treatment during ischemic stroke.

We needed to choose highly specific inhibitors for the transporter-specific inhibition study for the respective Oct transporters. Previous studies have shown that cationic substrate affinities for Octs often have overlapping inhibition curves, with less likely cationic compounds having affinities for either of the transporters [56,57]. However, a study showed that mitoxantrone is highly selective for the inhibition of Oct1 transporter because its estimated IC_50_ value for Oct1 is 40–60 fold lower than for Oct2 and Oct3 [53]. Thus, for our study, the inhibitory concentration for mitoxantrone was chosen at 25 µM, which is at least four fold greater than its IC_50_ value for Oct1. In this concentration, the respective transporter is inhibited by >80% (Table 1). Next, at 150 µM, corticosterone was selected for overall inhibition of Oct1, 2, and 3 because its IC_50_ value for either Octs was closely valued (Table 1). We further confirmed that the concentration of inhibitors was not toxic to bEnd.3 cells (Appendix A) and astrocytes (Appendix A). The permeability experiment for MF involved 30 min pre-incubation and incubation throughout 120 min with the inhibitors. The TEER values were obtained to ensure intact BBB after incubation with the inhibitors to corroborate with the cell viability study (Appendix A). With mitoxantrone, MF transport was significantly inhibited compared to the control (14.68 ± 0.9 × 10^−4^ cm/min, *p* < 0.05), which suggests that MF uses Oct1 for its transport (Figure 2. Additionally, with corticosterone, the inhibition was also significant compared to the control (12.86 ± 1.27 × 10^−4^ cm/min, *p* < 0.01), but not significantly different from the mitoxantrone treatment, which suggests that Oct1 is the primary transporter involved in MF’s transport across the BBB co-culture of bEnd.3 cells and astrocytes. 

Cationic transporters other than Oct1, 2 and 3, such as PMAT, MATE-1, MATE-2, and SERT, which have been shown to have affinities for MF [53,58] and expressed by the brain microvascular endothelial cells [58,59,60], were not explored for permeability studies. This was because concentration needed for all-cationic-mediated transporter inhibition (MPP+ at 5 mM) and PMAT inhibition (desipramine at 200 µM) were toxic to the bEnd.3 cells (Appendix A). Additionally, the use of non-toxic and highly selective inhibitors (based on their IC_50_ values) of PMAT, MATE-1, MATE-2, and SERT could be employed in future studies to identify their role in BBB transport of MF. 

We further explore MF’s transport during hypoxia using OGD conditions, as discussed below.

### 3.3. Permeability of MF during Oxygen Glucose Deprivation (OGD)

It is well documented that the paracellular permeability of molecules across BBB increases after 3–4 h of an ischemic stroke episode [61]. In fact, under hypoxia, brain endothelial cells have decreased protein expression of tight junctional proteins claudin-5, occludin, and ZO-1, which cause paracellular leak [62,63]. In addition to the paracellular leak, changes in transporters expression occur at the BBB following in vitro hypoxia or in vivo stroke modeling [34,36]. Such a change can drastically affect drug uptake during post-stroke recovery. To address this, we first exposed the co-culture model of BBB to 2, 4, and 6 h of OGD, examined the TEER values for the groups to have an idea of the integrity and viability of the barrier after hypoxia, and then evaluated the PC value of MF after quantification using the LC-MS/MS method. 

As shown in Figure 3A, compared to normoxia, there is no decrease in TEER following 2 h of OGD. However, with increased exposure of 4 h, we found a significant reduction in the TEER of the co-cultured barrier (*p* < 0.001), suggesting compromised integrity and, thus, a paracellular leak. Additionally, with 6 h of OGD, there is a further reduction in the barrier integrity compared to 4 h of OGD exposure (*p* < 0.01). These changes suggest that the barrier integrity is compromised with longer exposure time periods of 4 h and 6 h of hypoxia to the cells. Next, we determined the PC of MF during hypoxia and compared it with the normoxic control. Similar to TEER values, we found no significant difference with 2 h of OGD exposure in MF’s PC value (21.5 ± 2.2 × 10^−4^ cm/min). However, MF’s PC value significantly increased with either 4 or 6 h of OGD exposure (32.98 ± 2.27 × 10^−4^ cm/min and 33.82 ± 1.25 × 10^−4^ cm/min, *p* < 0.01) (Figure 3B). It is also worth noting that there isn’t a significant difference between 4 and 6 h of OGD exposure. However, our 6 h of TEER data showed a significant difference in barrier integrity loss compared to 4 h (Figure 3A). This made us further examine protein expression changes during the different OGD time points, as previous studies have shown differences in the regulation of transporters during such situations.

### 3.4. Protein Expression of Organic Cationic Transporters (Octs) in bEnd.3 Cells during Normoxia and Hypoxia 

Previous studies have shown that participation of endogenous transporters is crucial in transport of anionic and cationic drugs across the BBB [14,34,36,64]. It is thus important to understand the functional protein expression of transporters during disease conditions. Understanding this aspect will further accelerate or maximize neuroprotection by optimizing timing and dosage during ischemic stroke. A previous study from our lab showed that hypoxia caused increased expression of the OATP1 transporter, which resulted in increased permeability of biphalin [34]. However, limited studies have examined expression of Octs during ischemia. So, to determine that, first, we checked the expression of Oct1, 2, and 3 in bEnd.3 cells during normoxia. Consistent with previously reported studies that showed luminal expression of mainly Oct1 and Oct2 in human and rat brain endothelial cells, we found that our bEnd.3 cells stably expressed Oct1 and Oct2, however, with almost undetectable expression of Oct3 (Figure 4) [50,65,66]. This is also in alignment with our permeability data that showed Oct1 inhibition with mitoxantrone for MF’s permeability, thus further supporting Oct1’s involvement in MF’s transport. Additionally, as depicted in Figure 4, our protein expression experiment showed a stable Oct1 and Oct2 expression, but not Oct3 expression; thus, we conclude that Oct1 is the primary transporter responsible for MF’s transport across the in vitro co-culture BBB model. 

Next, we checked the expression of Octs during hypoxic time points of 2, 4, and 6 h. As depicted in Figure 4A, for Oct1 transporter, compared to the control, we did not find any change in expression with 2 h of OGD exposure. However, we found significant increase in expression at 4 h of OGD exposure compared to 2 h (*p* < 0.05). Interestingly, the expression was renormalized with 6 h of OGD exposure, similar to the control (Figure 4A). The significantly increased expression for 4 h OGD is consistent with our increased MF permeability result, indicating that the increased permeability during hypoxic conditions could be due to the enhanced Oct1 transporter expression. Although without significantly increased Oct1 expression for 6 h of exposure (compared to control), we found that the increases in the permeability of MF during ODG 4 h and 6 h were not dissimilar (Figure 3B). This could be due to a slight increase in Oct1 expression, but not significant, as shown by densitometric analysis (Figure 4A). It is also notable that the comparison in Oct1 expression for 4 h of OGD to 6 h of OGD had no significant difference (Figure 4A). Thus, carrier-mediated transport could play an important role in enhanced permeation of MF during longer 4 and 6 h of OGD exposure. Further experiments involving Octs inhibitors during longer OGD time-points exposure to the co-cultured cells were required to prove this assumption.

Next, for Oct2 transporter, we found no change in expression among the OGD time points and normoxic control (Figure 4B). Lastly, Oct3 transporter was almost undetectable in the tested cells during the OGD time points (Figure 4C).

### 3.5. Oct1 Involvement in MF Permeability during different Hypoxic Time Points Using OGD 

We wanted to confirm if the increased permeability of MF during 4 h of OGD exposure is due to an increase in Oct1 transporter expression. Thus, we performed permeability studies involving cells exposure to 4 h OGD in both mitoxantrone and corticosterone presence and absence. As shown in Figure 5A, the PC value of MF is reduced in the presence of inhibitors during the 4 h OGD exposure time point (from 27.58 ± 4.53 × 10^−4^ cm/min to 17.62 ± 1.35 × 10^−4^ cm/min using mitoxantrone and to 19.55 ± 2.96 × 10^−4^ cm/min using corticosterone, *p* < 0.05). This further suggests that Oct1 is a key determinant for MF transport during OGD and, thus, responsible for increased permeability of MF. It is also notable that there are no significant differences between inhibition while using mitoxantrone (Oct1 inhibitor) and corticosterone (Oct1, 2, and 3 inhibitor). It can also be extrapolated from Figure 5 that ~36% of MF transport during OGD is carrier-mediated (the inhibition component of transport during OGD is active transport), while the remaining ~64% is due to paracellular leakage. This result is almost similar compared to normoxia, where transport accounted for ~35% carrier-mediated and ~65% paracellular transport. Next, we performed permeability studies using inhibitors for 6 h of OGD exposure. As shown in Figure 5B, we found that the PC value of metformin significantly reduced in the presence of both inhibitors (from 30.71 ± 4.65 × 10^−4^ cm/min to 20.16 ± 0.11 × 10^−4^ cm/min using mitoxantrone and to 18.86 ± 0.85 × 10^−4^ cm/min using corticosterone, *p* < 0.05). This suggests that carrier-mediated transport is still a key determinant of MF transport during a longer exposure of 6 h OGD to the cells. 

The altered transporter expression (nutrients, ions, or drugs) at the BBB has been shown to be either beneficial or detrimental during ischemic stroke. A previous study reported that elevated expression of nutrient transporter GLUT1, which regulates glucose levels in the brain, served a beneficial role in reducing focal ischemia [67]. However, another nutrient transporter, SGLT, responsible for cell depolarization and glucose balance, causes increased edema formation when its expression is increased during ischemia [68]. Additionally, increased expression of ion transporter Na^+^-K^+^-Cl-cotransporter caused brain edema formation, while decreased expression of NA^+^/K^+^ ATPase caused accumulation of Na^+^, leading to endothelial swelling and cytotoxic edema [69,70]. Previously, our lab showed that functional expression of OATP1 is required for BBB transport of biphalin, during in vitro ischemia and reperfusion conditions [34]. Furthermore, we found that the expression of the transporter is increased, requiring consideration of the dosage and time window. Another study showed that the transport effect of Oct1 and Oct2 is critical to achieving therapeutic outcomes during in vivo ischemic stroke despite the paracellular leak [36]. 

Few scientific studies have examined the regulation of Octs. Hepatic nuclear factor 4α (HNF-4α) has been shown to regulate the gene expression of Octs, where they bind to the gene promotor regions of the transporters to modulate the mRNA expression [71]. Enhancing the cellular expression of HNF-4α has been shown to, in fact, increase Oct1 expression in human hepatocytes. Meanwhile, it is worth noting that HNF-4α has been shown to be identified as a transcription factor in brain tissues. Therefore, in future studies, checking the expression of HNF-4α along with Octs expression during various OGD time points will give us an idea of whether this transcription factor is responsible for the alteration in transporter expression during hypoxia in brain endothelial cells. Furthermore, our data suggest evolving expression of Oct1 transporter with a difference in ischemic exposure periods. Therefore, it is important to determine whether increased transporter expression is needed to reach MF’s therapeutic potential for ischemic stroke treatment. Future studies that involve inhibition of Octs and examining alteration in activation pathways such as NF-κB and mTOR, through which MF has been shown to mediate its neurotherapeutic actions, are warranted. Moreover, since we found increased MF permeability during hypoxia and that ~36% was mediated through active transport and the rest ~64% paracellular (Figure 5A), thus, it is warranted to check if this increase is critical to the activation of certain pathways involved in post-stroke recovery. In vivo behavioral ischemic stroke recovery studies routinely utilized in our lab could achieve this. 

### 3.6. Protein Expression of Octs in Primary Astrocytes during Normoxia and Hypoxia

Astrocytes play a key role in the maintenance and inductance of the BBB. Our lab previously showed that co-cultured in vitro BBB model of bEnd.3 cells and mouse primary astrocytes promoted barrier tightness compared to a monolayer BBB model of bEnD.3 cells [43]. Besides supporting barrier integrity, astrocytes have been shown to express and regulate various ions and drug transporters in the brain [72]. A previous study has reported immunoreactivity to Oct2, Oct3, and PMAT in some regions of mouse and rat brains [73]. Another study reported expression of Oct3 in primary human astrocytes [74]. Thus, we further wanted to check protein expression of Octs in mouse primary astrocytes (situated in our co-cultured model) during normoxia and different hypoxic time points. As seen in Figure 6, we found stable expression of Oct2 in the primary astrocytes without any statistically significant changes between normoxia and hypoxic OGD time points. We found no detectable expressions for Oct1 and Oct3 in our cultured mouse primary astrocytes, during both normoxia and hypoxia. 

In addition to the role of Oct1 transporter in bEnd.3 cells, the expression of Oct2 in astrocytes suggests that Oct2 might play a possible significant role in the transport/uptake of MF. Future studies involving MF uptake studies by primary astrocytes could be implemented to understand any significant uptake by the astrocytes. Additionally, uptake studies could be performed in the presence and absence of Oct inhibitors to explore their role in MF uptake in primary astrocytes. This will also help in determining if our reported high permeability value for MF is underestimated due to the amount of drug possibly being taken up by astrocytes, while getting transported from the apical to basolateral chamber of the in vitro co-culture BBB model. 

### 3.7. Interaction of Metformin with Plasma Glycoprotein (P-GP) Using P-GP Overexpressing Cell Line

P-GP is an ATP dependent efflux protein expressed in the apical compartment of the BBB, which restricts the movement of relatively large molecules (>400 Da) into the brain, by actively transporting the molecules back to the blood [75]. The assessment of UFR and ER using transwell assays with P-GP overexpressing MDR1-MDCK cell lines is commonly used to assess bidirectional movement of molecules through the cell monolayer [76]. This assay gives an idea of whether drug molecules have limitations in gaining access to the brain. Furthermore, studies have shown that permeability studies in MDR1-MDCK cell lines accurately reflect in vivo BBB transport [77,78]. For our study, we sought to investigate whether MF interacts with the P-GP transporter and thus predict the level of its P-GP transport potential. The unidirectional PC (from A > B and B > A) of MF, along with its UFR and ER are summarized in Table 1. We found the UFR and ER values for metformin to be 1.24 ± 0.08 and 0.63 ± 0.04, respectively (Table 2). The UFR of a compound increases depending upon the potential of being a P-GP substrate (compounds with UFR < 2 are considered as less potential substrates) and compounds that have an ER of >1.99 are considered as strong P-GP substrates [79,80,81]. Thus, with our findings, we conclude that MF does not interact with the efflux transporter P-GP in a P-GP overexpressing cell line. This result further bolsters our conclusion from transport experiments that MF should have access to the brain as it is highly permeable across the BBB during normoxia and hypoxia.

## 4. Conclusions and Future Studies

We conclude that MF is a highly permeable molecule across an in vitro co-culture BBB model. A key portion of its brain entry utilizes carrier-mediated active transport through Octs, while the remaining part involves passive paracellular diffusion. We found evolving expression patterns for Oct1 transport between hypoxic time periods and increased permeability for MF. Future studies should be conducted to decipher the exact mechanism causing the altered pattern of Oct1 transporter expression. In the future, MF brain pharmacokinetic studies will be completed for MF using an in vivo ischemic stroke model in our lab to determine the drug’s brain uptake clearance value (*Kin*) and correlate with our in vitro findings from this study. Octs inhibitors could be further used in vivo to decipher whether carrier-mediated transport is necessary to achieve therapeutic efficacy during ischemic stroke.

## Figures and Tables

**Figure 1 pharmaceutics-15-01357-f001:**
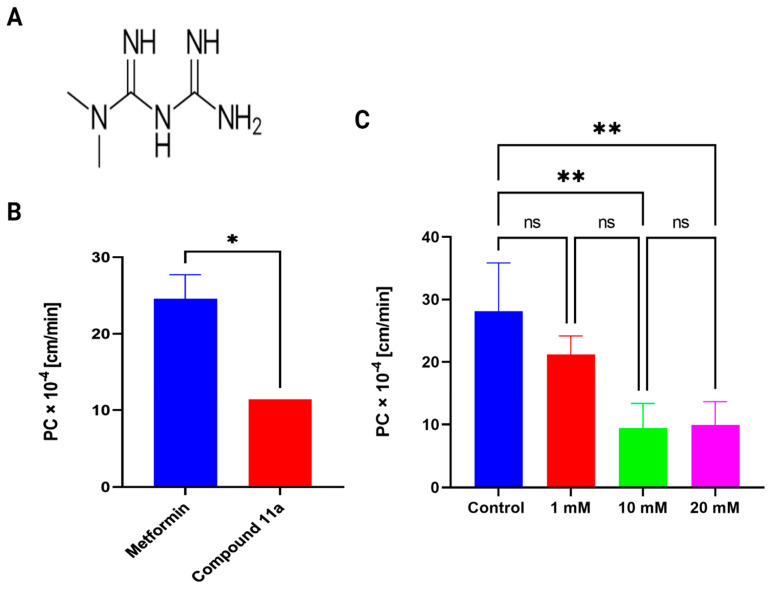
(**A**) Chemical structure of MF. (**B**) Permeability of MF across an in vitro co-culture setup of BBB. The permeability coefficient (PC) of MF compared to Compound **11a** [47], a neurolysin activator, experiments of which were performed under a similar setup of co-culture model of brain endothelial cells and astrocytes. Unpaired student’s *t*-test (* *p* < 0.05); *n* = 3 biological replicates; mean ± SD. (**C**) Saturable transport of tracer [^14^C] MF as shown by decrease in transport at self-inhibitory concentration of 10 mM, with no further significant decrease at 20 mM concentration. One-way ANOVA, followed by Tukey’s multiple comparisons test (** *p* < 0.01); ns = non-significant; *n* = 3 biological replicates; mean ± SD.

**Figure 2 pharmaceutics-15-01357-f002:**
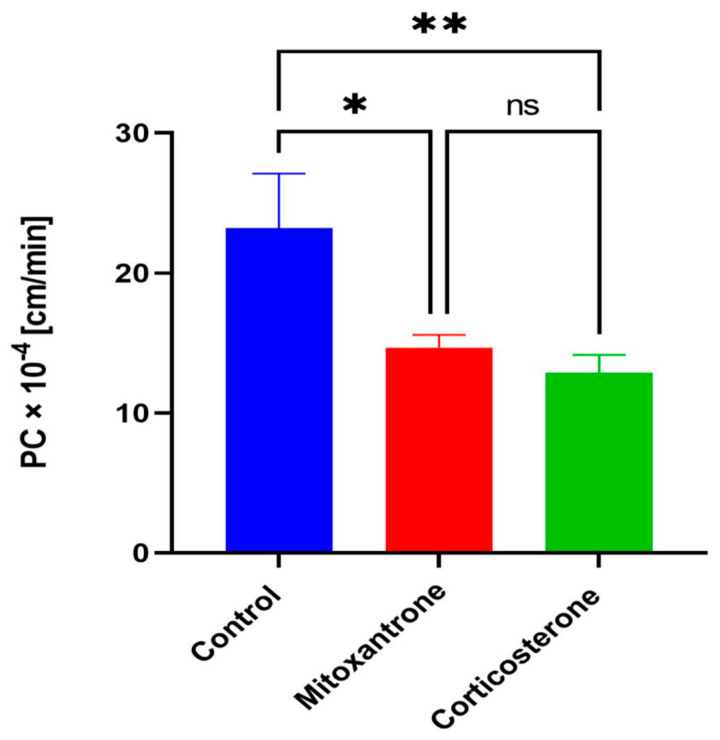
Permeability of MF using Octs specific transporter inhibitors. PC of MF significantly reduced with mitoxantrone (Oct1) and corticosterone (Oct1, 2 and 3) inhibitor. One-way ANOVA, followed by Tukey’s multiple comparisons test (* *p* < 0.05, ** *p* < 0.01); ns = non-significant; *n* = 3 biological replicates; mean ± SD.

**Figure 3 pharmaceutics-15-01357-f003:**
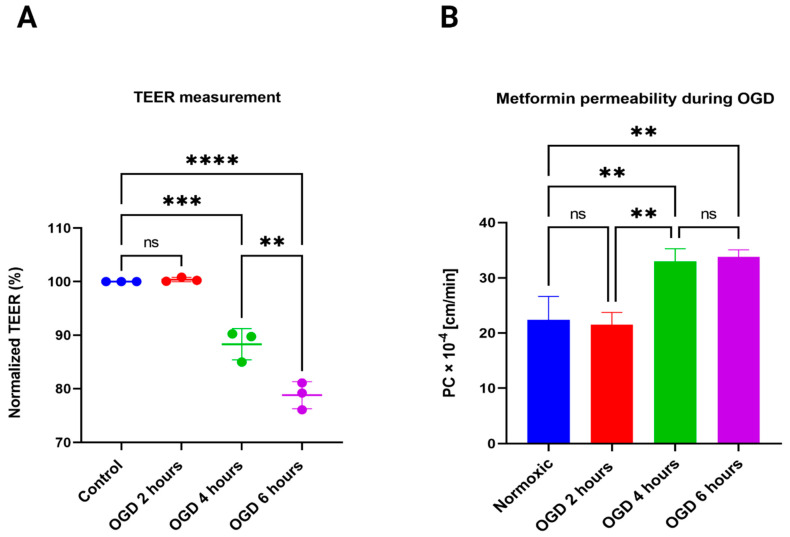
(**A**) OGD negatively impacts the BBB integrity, as demonstrated by TEER measurement. OGD exposure to the in vitro BBB model for 4 h significantly reduced the TEER compared to normoxia, and with 6 h exposure showed further reduction in TEER compared to 4 h exposure. One-way ANOVA, followed by Tukey’s multiple comparisons test (** *p* < 0.01) (*** *p* < 0.001) (**** *p* < 0.0001); *n* = 3 biological replicates; mean ± SD. (**B**) Permeability of MF during hypoxia. The PC of MF showed no change with 2 h exposure, however, with 4 h and 6 h exposure, there was a similar increase in permeability, compared to normoxia. No significant difference between 4 h and 6 h exposure with OGD. One-way ANOVA, followed by Tukey’s multiple comparisons test (** *p* < 0.01); ns = non-significant; *n* = 3 biological replicates; mean ± SD.

**Figure 4 pharmaceutics-15-01357-f004:**
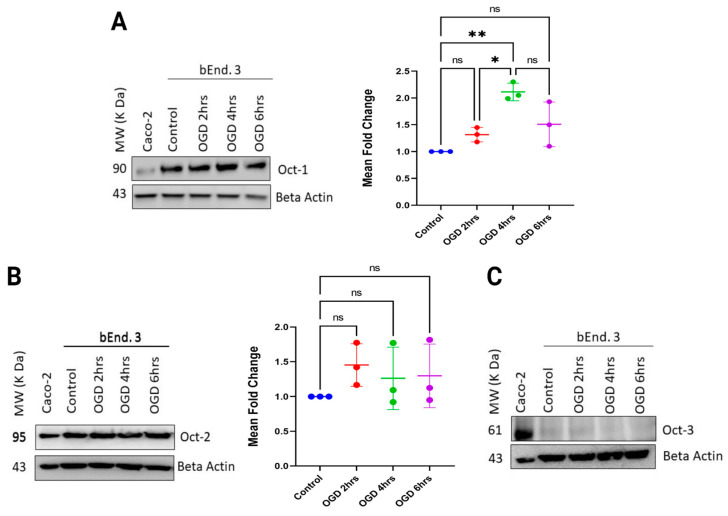
Densitometric analysis of organic cationic transporters (Octs) in bEnd.3 cells (immortalized mouse brain endothelial cells) during normoxia and hypoxia for 2, 4 and 6 h time points. (**A**) Expression of Oct1 in control cells (normoxia) and cells exposed for 2, 4 and, 6 h OGD (hypoxia), with evolving expression patterns seen among groups. (**B**) Stable Oct2 expression between the control and treatment groups, without significant differences in expression during OGD time-points. (**C**) Undetectable expression of Oct3 in control and treatment groups. The protein bands were quantified relative to beta-actin using Image J software and are reported as ratios of the control group. One-way ANOVA, followed by Tukey’s multiple comparisons test (* *p* < 0.05, ** *p* < 0.01); ns = non-significant; *n *= 3 biological replicates; mean ± SD.

**Figure 5 pharmaceutics-15-01357-f005:**
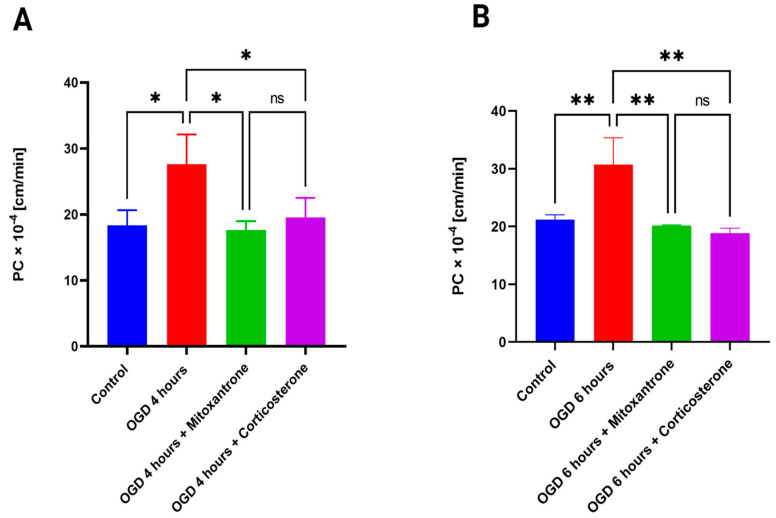
Permeability of MF after 4 h (**A**) and 6 h (**B**) of OGD exposure using mitoxantrone and corticosterone. The PC value of MF significantly reduces both with mitoxantrone and corticosterone in comparison to 4 and 6 h OGD exposure in absence of inhibitors. There is no significant difference in PC between both groups involving inhibitors after 4 and 6 h of OGD exposure. One-way ANOVA, followed by Tukey’s multiple comparisons test (* *p* < 0.05, ** *p* < 0.01); ns = non-significant; *n *= 3 biological replicates; mean ± SD.

**Figure 6 pharmaceutics-15-01357-f006:**
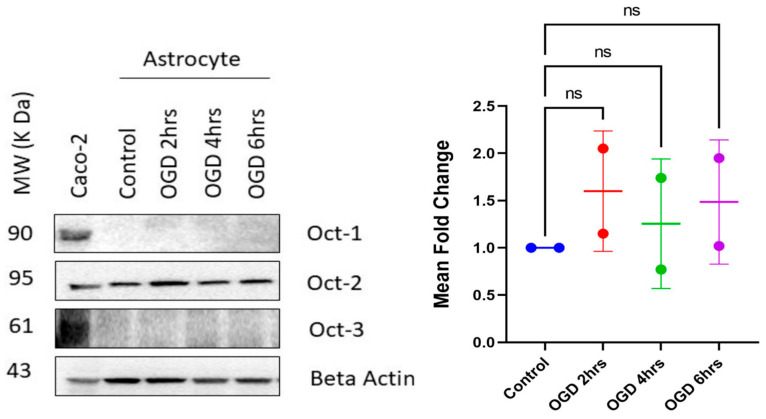
Densitometric analysis of organic cationic transporters (Octs) in mouse primary astrocytes during normoxia and hypoxia for 2, 4, and 6 h time points. Stable expression of Oct2 in control cells (normoxia) and cells exposed for 2, 4, and 6 h OGD (hypoxia), with no changes in expression patterns seen among groups. No detectable expression of Oct1 and Oct3 in control and treatment groups. The protein bands were quantified relative to beta-actin using Image J software and are reported as ratios of the control group. One-way ANOVA, followed by Tukey’s multiple comparisons test; ns = non-significant; *n* = 2 biological replicates; mean ± SD.

**Table 1 pharmaceutics-15-01357-t001:** Cited values from a literature for IC_50_ inhibition of MF uptake by organic cationic transporters (Octs) [53].

Inhibitors/Substrates	Specificity Based on IC50 Values for 10 µM Metformin Uptake		
	Oct-1	Oct-2	Oct-3
**Mitoxantrone**	3.0	135.0	174.0
**Corticosterone**	3.2	1.3	0.2

**Table 2 pharmaceutics-15-01357-t002:** MF’s PC in a P-GP overexpressing MDR1-MDCK cell line in presence and absence of a potent P-GP inhibitor, Cyclosporin A (CsA) at 10 µm to calculate the unidirectional flux ratio (UFR) and efflux ratio (ER). Mean ± SD of 4–5 replicates from 2 independent biological samples.

Metformin PC, A > B; 1 × 10^−7^ cm/s		Metformin PC, B > A; 1 × 10^−7^ cm/s	UFR (Unidirectional Flux Ratio)	ER (Efflux Ratio)
−CsA	+CsA			
2.58 ± 0.18	3.28 ± 0.06	1.63 ± 0.06	1.24 ± 0.08	0.63 ± 0.04

## Data Availability

Not applicable.

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
