# Peer review of "Permeability of Metformin across an In Vitro Blood–Brain Barrier Model during Normoxia and Oxygen-Glucose Deprivation Conditions: Role of Organic Cation Transporters (Octs)"

_pharmaceutics, 2023, doi:10.3390/pharmaceutics15051357_

Round 1

Reviewer 1 Report

The manuscript by Sharma and colleagues describes a preclinical study in which transport properties of metformin (MF), a highly prescribed drug used in the treatment of type II diabetes, in mouse brain microvascular endothelial cells were evaluated. The investigators propose a premise for this study that is based upon the potential utility of MF in ischemic stroke, which requires efficient blood-brain barrier (BBB) permeability. The investigators show that apical-to-basolateral flux of MF is primarily driven by Oct1 in cultured mouse brain endothelial cells (bEND.3). Interestingly, Oct-mediated MF uptake is enhanced following OGD treatment for 4 hours. Additionally, the investigators show that P-gp, a critical BBB efflux transporter, is not involved in the endothelial cell transport of MF. The studies described in this manuscript are well designed and the paper is well written. I do, however, have a few comments for the investigators that should be considered.

1. Integrity of the investigators' co-culture model was assessed, in part, via measurement of NaF permeability. This is problematic given that NaF is a transport substrate for Oat3 and Mrp2 (Hawkins et al. Neuroscience Lett. 2007; PMCID: PMC1785293). Since these transporters move their substrates in the basolateral-to-apical direction in vascular endothelial cells, use of NaF may result in an overestimation of endothelial cell layer integrity. Co-culture model integrity should be conducted using a small molecule tracer that is not a substrate for transporters known to be functionally expressed in brain micro vessel endothelial cells.

2. The concentration of [14C]MF used in the transport assays was reported to be 10 uM. Of significance, this concentration is in agreement with known plasma concentrations of MF in human subjects (i.e., steady-state concentrations of 1.8 ug/mL or approximately 13 uM following the maximum daily dose of 2,000 mg). The agreement between the concentration of MF used in the investigators' experiments and the established steady-state plasma concentrations in human subjects should be discussed in the manuscript. Additionally, how does the 10 uM concentration for [14C]MF relate to the Km value for Oct1, Oct2, and Oct3? This should also be discussed in the manuscript.

3. In the inhibition experiments, a concentration of 150 uM for corticosterone was used. Given that the IC50 for corticosterone against Oct isoforms does not exceed 3.2 uM (Figure 2A), why was such a large concentration used for this transport inhibitor? At this concentration, is corticosterone selective for Oct isoforms or could it potentially interact with other transporters in bEND.3 cells?

4. Figure 4 - a histogram showing densitometric analysis of protein levels measured via western blotting should be included. 

5. Perhaps the most important figure in the manuscript is figure 5, which indicates that selective transport by Oct isoforms is critical for MF permeability, even under an experimental condition that is known to increase paracellular diffusion (i.e., 4 h OGD). Of note, the investigators suggest that there any increase in MF uptake at 6 h OGD will be due to paracellular "leak" given that Oct1 protein levels revert to baseline under this condition (shown in Figure 4). This may not necessarily be true given that Figure 5 shows that selective transport is the primary determinant of membrane permeability of MF (Note the inhibitor data, which is not different from control). If paracellular leak was a key determinant of MF permeability under OGD conditions, the inhibitor groups would be higher than control but (potentially) lower than the OGD condition. As such, inclusion of comparable transport data at 6 h OGD should be added to strengthen the hypothesis that selective transport by Octs and not paracellular diffusion is the primary mechanism by which MF crosses the BBB in the setting of OGD.

6. Table 1 - data for B-A flux in the presence and absence of CsA should be included. This would greatly enhance the observation that P-gp is not a determinant of MF transport in bEND.3 cells.

7. Transporter Nomenclature - by convention, "OCT" refers to the human isoforms of organic cation transporters and "Oct" refers to rodent orthologues. Since the investigators are using murine cells, transporters in this model should be designated as "Oct1", "Oct2", and "Oct3" (not "OCT1", "OCT2", and "OCT3" as is done in the present manuscript). Please correct all transporter nomenclature.

8. In the introduction, it should be clarified that tPA is the only FDA-approved drug treatment for ischemic stroke. This delineation is important given that endovascular thrombectomy is also an approved treatment for stroke.

9. Title - should be modified to emphasize that transport studies were conducted under OGD conditions and not solely under hypoxic conditions. Perhaps the investigators will consider the following modification: "Permeability of metformin across an in vitro blood-brain barrier model during normoxia and oxygen/glucose deprivation conditions: Role of organic cation transporters (Octs)."

Reviewer 2 Report

The manuscript by Sharma and co-workers elucidates about the in vitro BBB transport of metformin (MF) using a mouse co-culture system and other relevant methodologies. It presents a very logical rationale behind the experiments that were conducted, which are supported by appropriate references and statistical analysis. Altogether, proving its likeability to cross the BBB, this works opens doors for further investigation on the potential of MF repurpusing for brain pathologies. Nevertheless, some minor concerns should be addressed and better clarified before publication:

- Line 76: typo (extra tab). Please correct.

- Section 3.1: It is missing a general introduction to the set of results present afterward, as main objectives (by similarity to following sections). Please include. Moreover, Fig. 1A is not mentioned in the text. Include.

- MF saturable transport: concentrations of 1, 10 and 20mM were employed. Are this pharmacologically relevant? And did the authors confirm they are not toxic to cells at these concentrations?

- Line 293: How was this extrapolation made from the data obtained? Same for lines 433-435. Please clarify in the main text.

- Section 3.3: It was observed MF permeability increased with 4 and 6h OGD. Are the values obtained comparable with other drugs similar to MF (e.g. regarding chemical structure or known to be transported by the same OCT)? Please include discussion.

- According to FDA, MF is described as an in vitro substrate of OCT2, MATE1, and MATE2 (https://www.fda.gov/drugs/drug-interactions-labeling/drug-development-and-drug-interactions-table-substrates-inhibitors-and-inducers#table4-1). Indeed, MATE1 and MATE2 expression has been reported in brain microvessels, both from human cell lines and from capillaries isolated from mice (as reviewed here https://www.ncbi.nlm.nih.gov/pmc/articles/PMC8836701/). Were these transporters’ contributions accounted? What is known regarding their expression in the cellular model employed? Please include discussion.

- Figure 4: What is the expression of the different OCT in astrocytes? And in the co-culture (b.End3 + astrocytes), as it is the experimental setting where previous experiments were made? It is well known that co-culture settings promote barrier tightness and the expression of several transporters at BBB level, different from b.End3 cultures alone. This evaluation of expression should be performed or at least discussed/accounted for the proposed mechanism of uptake of MF at BBB level.

- Figure 4: Include densitometric analysis graphs of 3 biological replicates of the WB, with statistics.

- Line 428 “This further suggests that the expression of OCT1 is increased during OGD and thus increases the permeability of MF.” Results from b.End3 cells WB during OGD do not corroborate such discussion proposed by the authors, since there were no changes in protein levels. Nevertheless, alterations in OCT1 activity cannot be discarded nor the contribution of others’ cells (since this is co-culture of b.End3 with astrocytes).  Please include.

- Table 1 location in the manuscript seems to be misplaced. Please correct.

- Section 3.6: P-gp is also expressed and highly abundant in HBMEC and at BBB level, and several authors report ABC transporters substrate accumulation assays using different HBMEC lines and substrates and inhibitors. Why did not the authors also tested P-gp involvement in b.End3 and in the co-culture model? It would be important to have a more comprehensive overview of the true mechanism of BBB transposition by MF.

Round 2

Reviewer 1 Report

The authors have addressed all concerns from the previous review. The manuscript is greatly improved and provides an important contribution to the field.